Comparative proteomic analysis for revealing the advantage mechanisms of salt-tolerant tomato (Solanum lycoperscium)

Wang Qiang 1 2
Wang Baike 2
Liu Huifang 2
http://orcid.org/0000-0001-7312-7924 Han Hongwei 2
Zhuang Hongmei 2
Wang Juan 2
Yang Tao 2
Wang Hao 2 wanghao183@163.com
Qin Yong 1 xjndqinyong@126.com
1 College of Horticulture, Xinjiang Agricultural University , Urumqi , China
2 Institute of Horticulture Crops, Xinjiang Academy of Agricultural Sciences , Urumqi , China
Nogueira Fabio
Electronic publication date: 2022 Feb 28
Publication date: 2022
Volume: 10
Electronic Location ID: e12955
Received 2021 Jul 29; Accepted 2022 Jan 27
Copyright: © 2022 Wang et al.
Copyright year: 2022
Copyright holder: Wang et al.
License: This is an open access article distributed under the terms of the Creative Commons Attribution License, which permits unrestricted use, distribution, reproduction and adaptation in any medium and for any purpose provided that it is properly attributed. For attribution, the original author(s), title, publication source (PeerJ) and either DOI or URL of the article must be cited.
License URL: https://creativecommons.org/licenses/by/4.0/

Keywords: Tomato (Solanum lycoperscium), Salt-tolerant, Proteome, Salt tolerant variety

Funding: National Natural Science Foundation of China 31860554, 31360482 This work was supported by the National Natural Science Foundation of China (No. 31860554; No. 31360482) The funders had no role in study design, data collection and analysis, decision to publish, or preparation of the manuscript.

==============================
Salt stress causes the quality change and significant yield loss of tomato. However, the resources of salt-resistant tomato were still deficient and the mechanisms of tomato resistance to salt stress were still unclear. In this study, the proteomic profiles of two salt-tolerant and salt-sensitive tomato cultivars were investigated to decipher the salt-resistance mechanism of tomato and provide novel resources for tomato breeding. We found high abundance proteins related to nitrate and amino acids metabolismsin the salt-tolerant cultivars. The significant increase in abundance of proteins involved in Brassinolides and GABA biosynthesis were verified in salt-tolerant cultivars, strengthening the salt resistance of tomato. Meanwhile, salt-tolerant cultivars with higher abundance and activity of antioxidant-related proteins have more advantages in dealing with reactive oxygen species caused by salt stress. Moreover, the salt-tolerant cultivars had higher photosynthetic activity based on overexpression of proteins functioned in chloroplast, guaranteeing the sufficient nutrient for plant growth under salt stress. Furthermore, three key proteins were identified as important salt-resistant resources for breeding salt-tolerant cultivars, including sterol side chain reductase, gamma aminobutyrate transaminase and starch synthase. Our results provided series valuable strategies for salt-tolerant cultivars which can be used in future.

Introduction

Soil salinization is a worldwide problem affecting agricultural production. With the destruction of soil caused by the expansion of the global industrial scale, nearly 20% of cultivated land and nearly half of the irrigated land are threatened by salt damage to varying degrees in the world. Soil salinization has become a major environmental problem which has been widely realized (Rozema & Flowers, 2008). The increasing trend of salinization poses a great threat to the sustainable development of agriculture. Currently, the agricultural improvement measures of salinization soil mainly include rational fertilization, increasing organic fertilizer and restraining soil salt super accumulation (Chen et al., 2011). However, the cultivation of saline-tolerant crop varieties is still an important method, which is meaningful to the improvement and utilization of salinized soil. As an important cash crop in the world, tomato growing was limited by various factors relevant to salt stress, resulting in a loss of yield. However, the salt-tolerance mechanism of tomato is still unclear and need further in-depth researches.

The major harmful ions in saline and alkaline soils is Na+. The accumulation of Na+ in soils could result in osmotic stress (Xiong & Zhu, 2002; Munns & Tester, 2008), ion toxicity (Tester & Davenport, 2003) and oxidative stress (Hasegawa, Bressan & Pardo, 2000; Xiong & Zhu, 2002), which profoundly influence the growth and development of plants. The excessive accumulation of Na+ in plant cells results in the destruction of the dynamic balance between the generation and clearance of oxygen free radicals, and leads to or intensifies the membrane lipid peroxidation and membrane lipid defatination (Senaratna, McKersie & Stinson, 1985), which is harmful to the normal physiological progresses of plants. Additionally, excessive Na+ decreases the concentration of Ca2+ in the plasma membrane, resulting in membrane structure destruction and changes in function, and intracellular exosmosis of potassium, phosphorus and organic solute (Tuna et al., 2007), which inhibited plant growth and development. It has been proved that salt stress can influence the utilization rate of starch, protein, fatty acids and other macromolecular substances in seeds, resulting in seed germination failure or delayed germination (Lovato, Martins & Lemos, 1994; Ungar, 1996). Meanwhile, the repair and reconstruction of cell membrane system of seeds were also impaired under saline-alkali stress. The membrane ion selective absorption capacity is decreased under salt stress, accompanying with ion toxicity and generation of free radicals which further damage the seed colloidal structure and slow down the development of seeds (Almansouri, Kinet & Lutts, 2001; Chartzoulakis & Loupassaki, 1997). Furthermore, salt stress could significantly influence the structure of plasma membrane. Salt stress can also lead the formation of large vacuoles, local expansion of endoplasmic reticulum, decrease in chloroplast numbers and enlargement of mitochondria (Mitsuya, Takeoka & Miyake, 2000; Bruns & Hecht-Buchholz, 1990).

Currently, various mechanisms of plant tolerance to salt stress have been elucidated, including reprogramming the photosynthetic pathway (Allakhverdiev et al., 2002; Mittal, Kumari & Sharma, 2012; Wang et al., 2021), enhancing the antioxidant protection ability (Gupta & Downer, 1993; Parvin et al., 2019; Faizan et al., 2021) and synthesis and accumulation of osmotic regulating substances (Yang et al., 2007; Ali et al., 2021). It has been reported that salt stress induce the increase in activity of SOD and APX, which plays important roles in the antioxidant protection system of plants (Zhang et al., 2014). Compared with wild-type tobacco, salt-resistant cultivars have higher photosynthetic efficiency and are more resistant to salt stress (Gupta & Downer, 1993). Under salt stress, plants usually need to synthesize and accumulate a large amount of solutes with active and non-toxic osmotic effects to reduce water potential (Chen et al., 2011), allowing cells to absorb water in a hypotonic environment. Moreover, plants also activate multiple signaling pathways to resist the harms caused by salt stress, including salt oversensitive (SOS) signal transduction pathway, jasmonic acid, mitogen-activated protein kinase (MAPK), and abscisic acid (ABA) signaling pathways (Xiong, Schumaker & Zhu, 2002; Zhu, 2002; Colcombet & Hirt, 2008). Therefore, plants regulate salt tolerance through a complex network of resistance response.

Tomato is a worldwide vegetable crop whose tolerance mechanism and proteomics of salt stress have been investigated extensively and deeply (Manaa et al., 2013; Gong et al., 2014). Comparative proteomic analyses are highly effective in the identification of protein changes that are associated with genotypic properties. For instance, an iTRAQ analysis of proteomes from maize inbred lines at salt-tolerant genotype F63 and the salt-sensitive genotype F35 showed that a large number of proteins was affected by salt in the two accessions (Cui et al., 2015). Those differentially expressed proteins may be helpful for further elucidating salt tolerance mechanisms.

The salt-tolerant introgression line IL7-5-5 was constructed by using 76 wild tomato Solanum Pennellii LA716 as donor and cultivated tomato M82 as acceptor. The salt-tolerant line IL7-5-5 was screened by repeated salt stress method at seedling stage. breeders observed that IL7-5-5 plants were salt-tolerant while M82 plants were susceptible to salt stress. In this study, we found that various proteins relevant to salt resistance were highly expressed in salt-tolerant cultivars, resulting in series advantages of ST in permeability regulation, maintaining normal photosynthesis and removing reactive oxygen species produced by salt stress. Our research provides a new blueprint for breeding salt tolerance tomato.

Materials and Methods

Plant material and experimental design

The experiment was carried out in the sunlight greenhouse of Xinjiang Academy of Agricultural Sciences in April 2019. The seeds of salt-tolerant infiltration system IL7-5-5 (ST) and salt-sensitive M82 (SS) were seeded in peat, perlite and vermiculite (volume ratio 1:1:1) substrates, seedling raising in conventional greenhouse. The roots were cleaned when the seedlings grow to the four-true-leaf stage, then transplanted the plant into 1/2 concentration Hoagland standard nutrient solution for hydroponic culture. The seedlings were cultivated at 25/18 °C (day/night) with a relative humidity of 40%, and at a photoperiod of 14 h/10 h (day/night). The oxygen was supplied to culture chambers three times everyday and 30 min every time. The evaporated water was replenished to its initial volume every 2 days, this culture method last for 1 week and the treatment started after this. The following experimental groups were arranged with three replicates and treated as follows: (1) CK: 1/2 Hoagland Nutrient Solution; (2) Salt Stress Treatment: 1/2 Hoagland nutrient solution with 200 mmol·L–1 NaCl. Sample the location of the same part of the material to be tested at 0, 1, and 12 h after treatment. The leaves were selected and stored in the refrigerator at –80 °C. All sample collection and preparation procedures were performed in triplicate, and each independent biological experiment has three technological replicates. Data were exhibited as the means ± standard deviations (SDs).

Determination of physiological indicators

Tomato leaves at 0 and 1 h after salt treatment were used for superoxide dismutase (SOD) activity, polyamine oxidase (PAO) activity, peroxidase (POD) activity, polyphenol oxidase (PPO) activity, soluble sugar content and H2O2 content assays by physiological trait kits (Suzhou Comin Biotechnology Co., Suzhou, China) according to the manufacturer’s instructions. A completely randomized block design was used, and all experiments had three replicates. Experimental data were analyzed by two-way analysis of variance (ANOVA) and Tukey (P < 0.05) comparisons were used to analyze differences.

Protein extraction, Trypsin digestion and isobaric labeling

Frozen tissue from tomato leaves was ground in liquid nitrogen using a mortar and pestle. Dry samples were mixed with lysis buffer (1% TritonX-100, 10 mM dithiothreitol, 1% protease inhibitor cocktail and 2 mM EDTA) and then processed three times on ice using a high intensity ultrasonic processor. After centrifugation (12,000×g, 4 °C, 10 min) using Tris-saturated phenol (pH 8.0) and transferring the supernatant, add at least four volumes of ammonium sulfate saturated methanol and allow the protein to precipitate overnight at 20 °C. Remove the supernatant by centrifugation (4 °C, 10 min) and wash three times with ice-cold acetone. Protein concentration was determined using the BCA kit. Protein was precipitated at 20% (m/v) TCA. Protein precipitates recovered after vortex mixing incubation (4 °C, 2 h) and centrifugation (4,500×g, 4 °C, 5 min) were washed with pre-chilled acetone, and the dried protein samples were re-dissolved in 200 nM TEAB and digested overnight with trypsin (1:50 w/w). Samples were reduced with 5 mM dithiothreitol (37 °C, 60 min) and alkylated with 11 mM iodoacetamide (room temperature, 45 min) in the dark. Peptide Isobaric labeling was performed using a TMT kit (Thermo Fisher Scientific, Waltham, MA, USA), with SS samples labeled with TMT tags 126, 127 and 128, and ST samples using TMT tags 129, 130 and 131.

Protein identification by LC-MS/MS

Separation of samples into fractions by high pH reversed-phase HPLC using Agilent 300 Extend C18 column (5 μm particles, 4.6 mm I.D., 250 mm length). The tryptic peptides were dissolved in solvent A (0.1% formic acid, 2% acetonitrile in water), then loading all the solution into a home-made reversed-phase analytical column. The whole operation was conducted on an EASY-nLC 1,000 Ultra Performance Liquid Chromatography system with a constant flow rate of 400 nl/min, and the detail information was followed: the gradient was comprised of an increase from 6% to 23% solvent in 98% acetonitrile over 26 min, 23% to 35% with 8 min and then climbing to 80% in 3 min, finally, hold at 80% for 3 min. The information of instrument applied during experiment was followed: electrospray voltage of 2.0 kV, m/z scan range of 350 to 1,800 for full scan, and intact peptides detecting resolution of 70,000. NCE was used to select peptides and the Orbitrap with a resolution of 17,500 was used to detect the fragments.

Data analysis

Maxquant search engine (vs 1.6.3.3, Cox & Mann, 2008) and uniprot database was use to process MS/MS data. Several protein types should be classified for accurate analysis and the detail information was followed: carbamidomethyl on Cys was recongnized as fixed modification and acetylation on protein N-terminal or oxidation on Met and deamidation (NQ) were classified as variable modifications. TMT-6plex quantification has also been performed. More software operation parameters were followed: mass tolerance for fragment ions was 0.02 Da, first search of mass tolerance for precursor ions was 20 ppm, main search of mass tolerance for precursor ions was 5 ppm, FDR was adjusted to <1% and minimum score for modified peptides was set >40.

Enrichment analysis

Gene ontology and KEGG pathway enrichment analyses were carried out by R packages cluster profiler. Proteins were classified by GO annotation into three categories: biological process, cellular compartment and molecular function. For each GO category, a two-tailed Fisher’s exact test was employed to test the enrichment of the differentially expressed protein against all identified proteins. Both pathway and GO terms with a P-value < 0.05 is considered significant. Furthermore, wolfpsort were used for a subcellular localization predication to predict the protein subcellular localization.

Protein–protein interaction network

All candidate proteins were searched against the STRING database version 10.1 for protein-protein interactions. Only interactions between the proteins belonging to the searched data set were selected, thereby excluding external candidates. Interaction network form STRING was visualized in Cytoscape 3.0. The settings used in STRING database was shown as below: Network type: full STRING network (the edges indicate both functional and physical protein associations); meaning of network edges: evidence; active interaction sources: Textmining, Experiments, Databases, Co-expression, Neighborhood, Gene Fusion and Co-occurrence.

qRT-PCR verification of gene expression of autophagy-related proteins

The expression of important salt related proteins (A0A0C6G3Q8, A0A3Q71RF5 and A0A37J2V8) was investigated by real-time quantitative polymerase chain reactions (RT-qPCR). Primer 6(v6.24) Designer was used to design the RT-qPCR specific primers (Table S1). The method of total RNA extraction, quantification and cDNA synthesis were described previously (Vieira et al., 2016). All experiments have been carried out at least three times with three independent biological replicates. The relative expression levels of target genes was calculated by the method of fold change of 2–ΔΔCt value (Schmittgen & Livak, 2008).

Results

Observable physicological changes between salt tolerant (ST) and salt sensitive (SS) cultivars

One hour after imposing, the content of carbohydrate, H2O2 and total protein in both salt-sensitive and salt-tolerant cultivars were determined. The results showed that no significant change was observed in the content of carbohydrate among both cultivars (Fig. 1A). Moreover, there was no significant difference in the content of H2O2 between in salt-tolerant tomato compared with the salt-sensitive tomato (Fig. 1B). One hour after imposing, the content of H2O2 in salt-sensitive tomato was still significantly higher than that in Salt-tolerant cultivars (Fig. 1B), which may cause damage to these cultivars. In parallel, there was no significant difference in content of total protein among both salt-sensitive and salt-tolerant cultivars. However, 1 h after imposing, the content of total protein in salt-tolerant cultivars was significantly increased to a higher level than salt-sensitive cultivars (Fig. 1C).

Figure 1 Physiological changes between salt-tolerant (ST) and salt-sensitive (SS) cultivars.

Basal content of carbohydrate (A), H2O2 (B) and total proteins (C) in SS and ST. An asterisk (*) represents significant difference at P < 0.05. Three biological replicates were performed.

Global proteome analysis of salt tolerant cultivars (ST) and salt sensitive (SS) cultivars

To further elucidate the mechanism of resistance to salt stress, mass spectroscopy based proteomics was performed. Both salt tolerant cultivar (ST) and salt sensitive (SS) cultivar treated after 0 h (ST and SS), 1 h (ST 1 h and SS 1 h) and 12 h (ST 12 h and SS 12 h) were harvested for proteomic analysis, and each treatment had three replicates. Furthermore, 7,663 function proteins were identified among all samples and analyzed by variation analysis (Table S2).

To investigate the influences of salt stress on cultivars (SS and ST), two algorithms were further performed, including principle components analysis (PCA) and unsupervised cluster analysis (Figs. 2A and 2B). First, PCA analysis showed an observable separation between all six treatments demonstrating that salt stress induced significant changes in proteomic profiles of both cultivars (SS and ST) and the responses to salt stress of ST were different to SS (Fig. 2B). Meanwhile, unsupervised cluster analysis showed a stability of replicates from each sample groups, validating our treatments (Fig. 2A). It further reach a consensus with PCA analysis that all replicates with same treatments clustered together and observably separated with each other. We further performed a two-factor (time point and strains) PCA analysis (Figs. 2C and 2D) on these proteomic data sets to detected the response to salt stress at different time point. Focus on the time points, we found that the most active responses were presented at 1 h after imposing (Fig. 2C). Interestingly, the significant changes at 0 h indicated that the basal level of proteomic profile between SS and ST were different with each other (Fig. 2D). Therefore, we proposed that the basal level of some salt resistance-related protein of ST were different with SS, resulting in a strong tolerance of ST to salt stress. Meanwhile, the ST can further make some responses under salt stress at 1 h, which is superior than SS in response to salt stress.

Figure 2 Landscape of proteomic profiles from all samples.

(A) Heat map showing the square of Pearson correlation coefficient between samples. The X and Y axes represent each sample. Coloring indicates the square of Pearson correlation coefficient (high: red; low: blue). (B) Principle component analysis displayed the separation of all treatments, and stability of replicates with the same treatment. Each circle represents a sample, the circles labeled by same color were the samples with the same treatment. (C and D) Two-factor principle component analysis showing the main variation of proteome distributed in samples after 1 h imposing, and comparisons of salt-tolerant (ST) and salt-sensitive (SS) cultivars.

Proteins located at chloroplast exhibited significant difference salt tolerant cultivar (ST) and salt sensitive (SS) cultivar

To investigate the differences in proteome between SS and ST before salt stress, we compared the proteomic profiles of ST and SS before treatment. Among 7,663 identified proteins, only protein in which the log1.5 (fold change) > 1 or < –1 and the P value < 0.05 were identified as differentially expressed proteins (DEPs; Fig. 3A). Finally, 278 proteins matched the condition were identified as DEPs (Fig. 3A). Among 278 DEPs, there are 131 up-regulated proteins and 147 down-regulated proteins in ST compared to SS (Fig. 3A). Stress-regulated genes, such as the lipoxygenase (A0A3Q7ENA4), superoxide dismutase (Q7XAV2) and gamma aminobutyrate transaminase 2 (A0A3Q7J2V8) were up-regulated in ST compared to their expression in SS, suggesting that ST have more advantages in resistance to salt stress than SS. Meanwhile, we found that the activity of two stress-related enzyme, SOD and PPO, were more active in ST than that in SS, whereas the enzyme of other two stress-related enzyme, POD and PAO, did not exhibited differences in activity in both SS and ST (Figs. 3B and 3C). We further analyze the location of these DEPs, the results indicated that most of differentially expressed proteins were located at chloroplast, cytoplasm, nucleus, extracellular, plasma membrane and mitochondria, demonstrating the proteins functioned at membrane and chloroplast contributed the differences of resistance to salt stress between SS and ST (Fig. 3E). It has been reported that various proteins at chloroplast and membrane play key role in resistance to salt stress, including superoxide dismutase (Q7XAV2) and gamma aminobutyrate transaminase 2 (A0A3Q7J2V8; Arora, Sairam & Srivastava, 2002; Blokhina, Virolainen & Fagerstedt, 2003). Therefore, we concluded that the basal resistance of ST to salt stress was higher than SS.

Figure 3 Location and variation of differentially expressed proteins between salt-tolerant (ST) and salt-sensitive (SS) cultivars.

(A) Volcano plots of differentially expressed proteins in SS and ST. Each plots represented a protein identified in SS and ST. The up-regulated proteins (Log1.5(Foldchange) >1.0; P < 0.05) were labeled by red, whereas the down-regulated proteins (Log1.5(Foldchange) < –1.0; P < 0.05) were shown in blue. (B–E) Basal enzyme activity of superoxide dismutase (SOD), polyphenol oxidase (PPO), peroxidase (POD) and polyamine oxidase (PAO) in SS and ST. Asterisk (*) indicates significantly different at P < 0.05. Three biological replicates were performed. (F) Location of differentially expressed proteins in ST vs SS pairwise comparison.

Gene enrichment analysis for DEGs from ST and SS

To study the differences of biological functions related to the diverse responses between SS and ST, we performed a gene ontology enrichment analysis on all 278 DEPs using TopGO (Fig. 4; Table S3). A total of 34 GO terms relevant to molecular progresses were significantly enriched based on these 278 DEPs, including heme binding, tetrapyrrole binding, peroxidase activity, oxidoreductase activity, peroxide acceptor, chitinase activity, antioxidant activity, intramolecular lyase activity and catalytic activity (Fig. 4A). Various terms were involved in the response to abiotic stress, such as peroxidase activity, oxidoreductase activity and chitinase activity. In addition, among identified cellular component-related terms, 20 terms were significantly enriched such as extracellular region, MCM complex, protein-DNA complex, replisome, nuclear replication fork and nuclear replisome (Fig. 4A). Under salt stress, a bunch of reactive oxygen species (ROS) could be produce in plants which cause damage to DNA and membrane (Gong et al., 2013; Singh & Bhatla, 2016; Scott, Meshnick & Eaton, 1987). Therefore, proteins having DNA-, ROS- and membrane-related function could protect the DNA from damage, ensuring normal biological processes. Finally, compared to SS, the results of GO enrichment analysis further showed that 125 terms relevant to biological process category were significantly enriched (P < 0.05; Fig. 4A). Remarkably, many stress-related biological processes were significantly identified, including response to stress, reactive oxygen species metabolic process, hydrogen peroxide metabolic process, response to stimulus, chitin metabolic process, cell wall macromolecule catabolic process, response to oxidative stress, response to external stimulus and cellular carbohydrate metabolic process (Fig. 4A), indicating that the resistance response to salt stress were different between SS and ST. Moreover, the results of KOG analysis on these DEPs showed that the expression of proteins which involved in signal transduction mechanisms, energy production and conversion, carbohydrate transport and metabolism, amino acid transport and metabolism, lipid transport and metabolism, secondary metabolites biosynthesis, transport and catabolism were significantly altered in ST compared to SS (Fig. 4B). It has been proved that amino acids such as proline and lipids such as glyceride play important functions in salt resistance of plants (Ashraf & Foolad, 2007; Butt et al., 2016; Huang et al., 2014). In parallel, signal transduction also influence the plant salt resistance, such as hormone JA and ABA (Msanne et al., 2011; Müller & Munné-Bosch, 2015; Zhu et al., 2010; Li et al., 2015; Huang et al., 2012; Guo et al., 2015). Various studies have proved the important function of JA and ABA in strong resistance responses of plants to salt stress (Ding et al., 2016; Me, Zhang & Huang, 2016; Islam et al., 2017; Jiang et al., 2016; Liu et al., 2015b; Gurmani et al., 2013). In the pathways analysis, various biosynthesis relevant to JA and ABA were significantly enriched, including α-linoleic acid biosynthesis and steroid biosynthesis (Fig. 4A). Based on these results above, we proposed that ST have more advantages in resisting salt stress compared to SS.

Figure 4 Function enrichment analysis on all differentially expressed proteins in salt-tolerant (ST) vs salt-sensitive (SS) cultivars.

(A) All differentially expressed proteins with Gene Ontology matches were assigned to three GO categories (cellular component, molecular function, and biological process). The terms relevant to biological progress was labeled by yellow square, where as the cellular component and molecular function were showed in blue and green squares, respectively. The number of proteins in each term was shown behind the square. (B) Sequences with Gene Ontology matches were assigned to the KOG database and classified into 24 functional categories. The number of proteins in each categorizes was shown on the top the each column.

Proteins involved in various salt resistance-related pathways were altered in ST

Similarly, pathway enrichment analysis were also performed on all DEPs against KEGG database. The results showed that there were 84 biosynthesis pathways were enriched (Fig. 5A). Among 84 pathways, 14 pathways were significantly enriched (Fig. 5A; P < 0.05; Table S4), including phenylpropanoid biosynthesis, metabolic pathways, flavonoid biosynthesis, biosynthesis of secondary metabolites, DNA replication, nitrogen metabolism, valine, leucine and isoleucine degradation and zeatin biosynthesis (Fig. 5A). It has been reported that secondary metabolites played important roles in response to salt stress (Hazman et al., 2015). The increase in content of betaine and sorbital could enhance the resistance of various varieties to salt stress (Orthen, Popp & Smirnoff, 1994; Kanayama et al., 2007; Gao et al., 2000; Fu et al., 2011). Remarkably, all these metabolites-related pathways were significantly reprogrammed in ST (Fig. 5A), which may contribute to the advantages of ST in response to salt stress. Based on these three function analyses, 112 DEPs which functioned in these terms were identified (Fig. 5B). In the expression pattern of these DEPs, two modules were constructed in the heatmap (Fig. 5B). Among these 112 DEPs, 76 were up-regulated in ST and clustered at the top module which was labeled by pink square, whereas 36 proteins were down-regulated (Fig. 5B). Among highly up-regulated proteins, Superoxide dismutase (Q7XAV2), Lipoxygenase (A0A3Q7ENA4) and various proteins relevant to chloroplast were involved in plant resistance response to salt stress (Smirnoff, 2005; Andronis & Roubelakis-Angelakis, 2010; Schreiner & Zozor, 1996). All of the 76 up-regulated DEPs were further recognized as key proteins which were important for the account for the mechanism of stronger salt resistance of ST than SS.

Figure 5 Pathway enrichment analysis of differentially expressed proteins in salt-tolerant (ST) vs salt-sensitive (SS) cultivars.

(A) Top 20 of Enriched pathways of differentially expressed proteins; The size of circle represented the number of proteins in the enriched pathway. And the significance were shown in different color (high: blue; low: red; P < 0.05). (B) Expression pattern of differentially expressed proteins in ST and SS. The highly expressed proteins in ST were shown in red, and the down-regulated proteins in ST were labeled by blue. Each cell represented a differentially expressed protein, and the columns were the samples of SS and ST.

Responses of ST and SS were different under salt stress

To further clarify the advantage of ST in response to salt stress, we compared the proteomic profiles of ST and SS after 1 h imposing, which was observably separated with each other in PCA plots. In ST1 vs SS1 pairwise comparison, 169 proteins were identified as DEPs (Fig. 6A). Among 169 DEPs, 49 proteins were up-regulated in ST, whereas 120 proteins were up-regulated in SS (Fig. 6A). The expression level of various proteins functioned in response to salt stress were higher than SS, including Photosystem II reaction center protein, non-specific lipid-transfer protein 2-like, ABC transporter F family member 5, anion: sodium symporter and gamma aminobutyrate transaminase 2. It has been proved that high expression level of lipid-transfer protein could enhance the resistance of plants to salt stress, and such protein could also be induced to up-regulated under salt stress (Li, Yang & Zhang, 2016; Torres-Schumann, Godoy & Pintor-Toro, 1992). Interestingly, all the DEPs were still mainly located at chloroplast, nucleus, cytoplasm, extracellular and plasma membrane (Fig. 6E). Remarkably, after 1 h imposing, the enzyme activity of POD, PAO and PPO were further activated in ST, whereas the activity of these three stress-related enzyme were decreased in SS, which account for the advantage of ST in tolerance of salt challenge (Table 1). However, the activity of SOD did not display significant changes in SS following salt stress challenges (Table 1). Overall, all the 49 up-regulated DEPs were recognized as focus for further analysis below to clarify the advantage of ST.

Figure 6 Location and variation of differentially expressed proteins between salt-tolerant (ST) and salt-sensitive (SS) cultivars after 1 h imposing.

(A) Volcano plots of differentially expressed proteins in SS1 and ST1. Each plots represented a protein identified in SS and ST. The up-regulated proteins (Log1.5(Foldchange) >1.0; P < 0.05) were labeled by blue, whereas the down-regulated proteins (Log1.5(Foldchange) < –1.0; P < 0.05) were shown in red. The green points were the proteins without significant expression. (B) Location of differentially expressed proteins in ST1 vs SS1 pairwise comparison. Each column represented a structure of organisms.

Table 1 Enzyme activity of salt-tolerant (ST) and salt-sensitive (SS) cultivars under salt stress challenge.

Enzyme Activity (U/g)	
	POD	PAO	PPO	SOD	
SS0	94.99 ± 3.15b	6.75 ± 0.82b	90.53 ± 4.43bc	1,080.44 ± 20.60c	
SS1	49.65 ± 1.14d	4.30 ± 0.34c	81.28 ± 5.33c	1,045.90 ± 9.24c	
ST0	90.22 ± 1.02c	6.30 ± 0.33b	96.57 ± 6.41b	1,197.42 ± 28.38a	
ST1	145.98 ± 1.77a	8.67 ± 0.79a	110.52 ± 4.65a	1,117.93 ± 13.24b	
Note:

Significantly different at P < 0.05. Three biological replicates were performed.

The advantage of ST in response to salt stress

To elucidate the advantage of ST in response to salt stress after 1 h imposing, we further performed function analysis. Compared to SS, all the up-regulated DEPs in ST after 1 h imposing were analyzed by Gene Ontology enrichment analysis (Figs. 7A–7C). The results showed that many significantly enriched molecular function-related terms were involved in salt stress, which were helpful for plants to tolerate salt stress, including electron transporter, transferring electrons within the cyclic electron transport, pathway of photosynthesis activity, structural constituent of cytoskeleton, 4-aminobutyrate transaminase activity, ATP:ADP antiporter activity, heat shock protein binding (Fig. 7B). Enzymes with ATPase activity perform important roles in plant resistance to salt stress (Rangani et al., 2016). Meanwhile, aminobutyric acid is also an important chemical component in plants response to salt stress, and 4-Aminobutanoate (GABA) can alleviate the influences of salt stress on a variety of plants, including tomato, corn and wheat (Li et al., 2016; Luo et al., 2012; Shi et al., 2010). For cellular component categorizes, gene functioned in chloroplast-related terms were highly up-regulated in ST under salt stress, including chloroplast, chloroplast part and chloroplast thylakoid (Fig. 7A). Under salt stress, chloroplast could help plant scavenge the excessive accumulation of ROS and maintain the normal growth of plants (Li et al., 2016; Lu, Duan & Li, 2007). Furthermore, the significantly enriched biological progresses relevant to salt stress were response to wounding, cellular carbohydrate metabolic process, phloem development, cell wall biogenesis, response to stimulus and response to hydrogen peroxide (Fig. 7C). The strengthened cell wall and carbohydrate metabolites are important for plants as being challenged by salt stress (Pang et al., 2016; Dinneny, 2015; Liu et al., 2015a; Khan, Siddique & Colmer, 2016). Further pathway analysis on 169 DEPs showed an overrepresentation of pathways relevant to resistance to salt stress, such as Alanine, aspartate and glutamate metabolism, alpha-Linolenic acid metabolism, Linoleic acid metabolism, Peroxisome, Steroid biosynthesis, Valine, leucine and isoleucine degradation (Fig. 7D). Amino acid proline could contribute the tolerance of plants to salt stress by functioning as a protective agent and nitrogen source for cell structure and enzymes (Mansour, 1998; Butt et al., 2016). Meanwhile, steroid biosynthesis could increase the content of ABA, which was widely identified in plant response to salt stress and played important function in these processes (Albacete et al., 2008, 2010). Remarkably, 21 proteins were identified from these important pathways and GO function terms, including Lipoxygenase (Q84P54), gamma aminobutyrate transaminase 2 (A0A3Q7H8R6), Peroxidase (A0A3Q7EK65) and Glutamine synthetase (A0A3Q7J2V8) (Fig. 7E). Protein Peroxidase have been widely recognized as important role in salt resistance. Therefore, we proposed that the advantage of ST in response to salt stress was mainly activation of salt resistance-related metabolisms and alteration of cell structure.

Figure 7 Function and pathway enrichment analysis on up-regulated proteins in salt-tolerant (ST) after 1 h imposing.

Top 20 of Enriched Gene ontology terms of differentially expressed proteins in ST after 1 h imposing; The size of circle represented the number of proteins in the enriched GO terms. The significance was shown in different colors (high: blue; low: red; P < 0.05); (A) cellular component categorization; (B) molecular function categorize; (C) biological progress categorization. (D) top 20 of enriched pathways of differentially expressed proteins in ST following 1 h salt stress; The length of column represented the number of proteins in the enriched pathway. And the significance were shown in different color (high: blue; low: black; P < 0.05). (E) Expression pattern of differentially expressed proteins in ST and SS following 1 h salt stress. The highly expressed proteins which involved in response to salt stress in ST were shown in red, and the down-regulated proteins in ST were labeled by green. Each cell represented a differentially expressed protein, and the columns were the samples of SS and ST after 1 h imposing.

Three key function proteins were important for the salt tolerance of ST

Overall, we found that three up-regulated proteins were shared in both ST vs SS and ST1 vs SS1 pairwise comparisons, including A0A0C6G3Q8, A0A3Q7J2V8 and A0A3Q7IRF5 (Fig. 8A; Table 2). Both A0A3Q7J2V8 and A0A3Q7IRF5 perform function at chloroplast, whereas A0A0C6G3Q8 is located at plasma membrane (Table 2). Remarkably, all these three proteins up-regulated 1.8~2.0-fold in ST vs SS pairwise comparison (Figs. 8C–8E). And their expression in SS following 1 h imposing was still almost 2.0-fold lower than ST after 1 h imposing (Figs. 8C–8E). However, all of them could be induced to up-regulated in SS after 1 h imposing (Figs. 8C–8E). By function analysis of each proteins, we found that A0A0C6G3Q8 has Sterol side chain reductase function, and A0A3Q7J2V8 is gamma aminobutyrate transaminase, whereas A0A3Q7IRF5 involved in Starch synthase (Table 2). Remarkably, up-regulation of A0A0C6G3Q8 could promote the synthesis of Brassinolide in vivo, and the increase in content of brassinolide may also contribute the plant resistance to salt stress and freezing injury, which has been widely used in agriculture (Divi, Rahman & Krishna, 2010; Xia et al., 2009; Ye, Li & Yin, 2011). Additionally, up-regulation of A0A3Q7J2V8 encoding gamma aminobutyrate transaminase promotes synthesis of GABA which has been widely used in filed to enhance plant resistance to various abiotic stress (Li et al., 2016; Shi et al., 2010). Importantly, in the co-expression network we further observed the hub role of these three proteins that all of them were located at the core of the network and closely correlated with other up-regulated proteins which involved in resistance to salt stress (Fig. 8B). qRT-PCR analysis was further used for validation of these three key proteins of ST in response to salt stress. The results showed there was no significant differences in basal expression level of these three genes in both SS and ST cultivars. In contrast, dramatic up-regulation of these genes was induced by 1 h imposing in ST cultivars, whereas their expression levels were not altered in SS after 1 h imposing, leading a strong tolerance of ST to salt stress (Fig. 9). Therefore, we highlight the importance of A0A0C6G3Q8, A0A3Q7J2V8 and A0A3Q7IRF5 in the mechanism of resistance of ST to salt stress.

Figure 8 Three key proteins were involved in mechanism network of resistance of salt-tolerant (ST) to salt stress.

(A) Venn diagram of up-regulated proteins in ST vs SS and ST1 vs SS1. Three up-regulated proteins were commonly identified in ST and ST1 after 1 h imposing. (B) Co-expression network of all up-regulated proteins which relevant to salt stress in ST. The key genes in the network were shown in red circles, indicating their hub status in response to salt stress. (C–E) Relative expression of key proteins in each pairwise comparisons. The column with different color represented a singly pairwise comparison. Asterisk (*) represents significantly different at P < 0.05.

Table 2 Key proteins of salt-tolerant (ST) in response to salt stress.

Protein accession	Protein description	Subcellular Localization	KEGG	
A0A0C6G3Q8	Sterol side chain reductase	plasma membrane	K09828	
A0A3Q7J2V8	gamma aminobutyrate transaminase 2	chloroplast	K16871	
A0A3Q7IRF5	Starch synthase, chloroplastic/amyloplastic	chloroplast	K13679	
Note:

P < 0.05.

Figure 9 Validation of key genes expression using qRT-PCR.

The relative expression level was represented by the 2–ΔΔCt value, and the SS and ST were shown in green and red columns. Asterisks (**) indicate significantly different at P < 0.01.

Discussion

Salt stress seriously influences growth and development of tomato, which seriously restricts the yield of tomato. With the development of researches, it has been showed that plants can resist salt stress by various physiological changes. Among our both ST and SS cultivars, the activity of SOD, PPO and POD was higher in ST cultivars, leading a strong tolerance to salt stress. Subsequently, in this study, the changes in proteomic profiles of salt-tolerant and salt-sensitive tomato varieties were compared. We found that salt-tolerant cultivars and salt-sensitive cultivars had significantly different responses to salt stress.

Salt-tolerant cultivars could enhance the tolerance of plants to salt stress by changing the expression of membrane structure-related proteins. Meanwhile, a variety of metabolic pathways were altered between salt-tolerant and salt-sensitive cultivars. Compared with the salt-sensitive cultivars, there is a higher active pattern of metabolisms relevant to Nitrate and amino acids. And under the challenge of salt stress, the salt-tolerant cultivars can further activated the Brassinolides and GABA biosynthesis, which contribute to the resistance to salt stress. Importantly, the higher expression of chloroplast-related protein in salt-tolerant cultivars indicated that salt-tolerant cultivars had higher photosynthetic activity, which could promotenormal growth of plants under salt stress. Additionally, salt-tolerant plants have advantages in scavenging reactive oxygen species produced by salt stress. A bunch of antioxidant-related proteins were activated to up-regulated in salt-tolerant cultivars, such as peroxidase and Catalase isozyme. In-depth analysis, we identified three key proteins, which highly expressed in salt-tolerant cultivars and play key role in its strong resistance to salt stress, including sterol side chain reductase, gamma aminobutyrate transaminase and starch synthase.

N-related metabolisms play important function in salt-tolerance of plants. A remarkable property of salt-tolerant is that the proteins involved in N metabolism were highly up-regulated in ST. It has been documented that high value of pH caused by salt stress will significantly influence the N metabolism and absorption of plants (Groppa & Benavides, 2008; Shan et al., 2016). The absorption of NO3– by plants is mainly regulated by H+/NO3– co-transporters, which requires high concentration of protons around the channel as the transport energy (Dluzniewska et al., 2007). Under the salt stress, the high pH environment around the roots leads to the shortage of protons around the channels, which greatly restricts the assimilation or absorption of NO3– by the roots, and inhibits the transport of NO3–, resulting in the decrease of the content of NO3– in the roots, thus interfering the whole nitrogen metabolism pathway (Abouelsaad, Weihrauch & Renault, 2016; Murtaza et al., 2013; Shao et al., 2015). Our proteomic profiles of Salt-tolerant and Salt-sensitive cultivars revealed that various proteins involved in N metabolism were highly up-regulated in salt-tolerant cultivars, in contrast these proteins were down-regulated in salt-sensitive cultivars such as AMT and NRT. Some studies have shown that plant roots transport NH4+ through AMT protein family, and transport NO3– by NRT protein family. Under salt stress, the expression levels of OsNRT and most members of OsAMT family in rice roots are increased, for the increase in absorption of NO3– and NH4+ to make up for the deficiency of content in roots. These results suggested that the N metabolism were not impaired in salt-tolerant cultivars, which maintains the normal physiological requirements of plants and enhance the tolerance of plants to salt stress. Therefore, one of the advantages of salt-tolerant cultivars was relative high levels of N metabolism. Similarly, the salt-tolerant rice were also exhibited high level of N metabolism, which contribute to its tolerance to salt stress (Lutts, Majerus & Kinet, 1999). Meanwhile, it also has been proved that the amino acids-, carbohydrate- and flavonoids-related biosynthesis also performed important function in the adaption of plants to salt stress.

Subsequently, our proteomic data showed a high expression level of genes relevant to GABA and brassinolides biosynthesis. BR can improve plant resistance to biological and abiotic stress (Divi, Rahman & Krishna, 2010; Ye, Li & Yin, 2011). The regulation of BR on plant stress resistance was closely related to the accumulation of H2O2 and the enhancement of the activity of antioxidant enzymes (SOD, CAT, APX; Xia et al., 2009; Zhang et al., 2010). Exogenous BR can significantly improve the growth and development of seedling plants under salt stress, reduce salt damage index, and significantly increase the content of proline and soluble sugar in leaf cells and the activity of antioxidant enzymes such as SOD, POD and CAT (Shang et al., 2006), thus improving the salt resistance of cucumber seedlings. Meanwhile, exogenous BR can improve the salt tolerance of Arabidopsis thaliana (Ye, Li & Yin, 2011). Under salt stress, exogenous EBR could promote the activities of CAT and SOD enzymes in tomatoes, reduce the content of MDA in cells, and improve the salt tolerance of tomatoes (Ahammed et al., 2013). Gamma-aminobutyric acid also plays an important role in plant salt tolerance, GABA can regulate cytoplasmic pH, resist oxidative stress, interact with microorganisms, and defend insects as osmotic regulator and signal molecule (Gilliham & Tyerman, 2016; Bouché & Fromm, 2004). Appropriate concentration of GABA can effectively promote seed germination and seedling growth under salt stress (Ziogas et al., 2017). GABA can reduce oxidative damage by improving the antioxidant capacity of plants (Li et al., 2016). At the same time, it can improve the content of osmotic regulation substances, maintain osmotic balance (Li et al., 2010), inhibit photosynthetic pigment degradation, and maintain photosystem II (PSII) activity. In addition, GABA can further regulate plant hormone content (Shi et al., 2010) to improve plant salt tolerance. Therefore, high basal level of BRs and GABA biosynthesis will be a new direction for breeding salt-tolerant cultivars.

Chloroplast were also important for the tolerance of plants to salt stress. Under abiotic stress, the most vulnerable part is PSII (Wei et al., 2012). The photosynthetic rate decreased due to salt stress, and salt stress would also reduce leaf turgor pressure and cause partial stomatal closure to further inhibit photosynthesis (Chaves, Flexas & Pinheiro, 2009). Our data showed a high expression pattern of proteins which involved in chloroplast membrane, demonstrating that the Salt-tolerant cultivars have advantages in response to salt stress challenge. The salt-tolerant cultivars showed an overrepresentation of chloroplast-related terms, which indicated that a bunch of proteins functioned in chloroplast were purposefully highly expressed. The reason is that plants need to maintain the photosynthesis of plants at normal level, which providing adequate nutrient for plants to enhance tolerance to salt stress. Additionally, various hormone-related biosynthesis were also closely correlated with the chloroplast, such as ABA, JA and GA3. Meanwhile, plant hormones play important roles in the signal transduction of plants in response to salt-stress, including ABA and BRs (Yang et al., 2011; Dodd et al., 2009). Hormone synthesis is induced by various abiotic stress, and their content varies with the degree of salt-stress. The changes in content of hormone will initiate or regulate some physiological and biochemical processes relevant to the response to abiotic stress. Researches have indicated that exogenous application of ABA will enhance the tolerance of plants to salt stress (Dodd et al., 2009). We noted that both biosynthesis of ABA and BRs were activated in Salt-tolerant cultivars, whereas these biosynthesis exhibited an low level of activity in Salt-sensitive cultivars.

The ROS scavenging system is imperative for plant growth and development (Gémes et al., 2011; Hajiboland et al., 2010). Under salt stress challenge, Plants always improve their salt-tolerance by regulating the dynamic balance of reactive oxygen species (ROS), which is caused by salt stress and results in damage to plants. In our research, we found that proteins involved in ROS scavenging system was highly expressed in salt-tolerant tomato, suggesting that the salt-tolerant cultivars have more advantages in scavenging of ROS, whose accumulation was caused by salt stress, than salt-sensitive cultivars. It has been proved that excessive accumulation of reactive oxygen species will damage the integrity of the plant cell membrane system and influence the plants development and growth (Smirnoff, 1993; Asada, 1999; Shigeoka et al., 2002). Remarkably, excessive ROS accumulation will also cause damage to the ultrastructure of chloroplast, which influences the photosynthetic system (Gémes et al., 2011). In parallel, it also could inhibit the activity of photosynthesis-related enzyme, leading a decrease in efficient of photosynthetic system (Gémes et al., 2011). In contrast, we have provided full evidences that the chloroplast and photosynthesis of salt-tolerant cultivars were more active than that in salt-sensitive cultivars. Active photosynthesis also could contribute to the scavenging of ROS by activation of SOD (Arora, Sairam & Srivastava, 2002; Andronis & Roubelakis-Angelakis, 2010). Similar to our results, it has been proved that there is less accumulation of ROS in the salt-tolerant cultivars, with high photosynthesis in vivo. Therefore, we further recognized the highly active ROS scavenging system as an important advantage of Salt-tolerant tomato as being challenges by salt stress.

Soil salinization has greatly restricted the production and development of agriculture (Gupta & Huang, 2014). Breeding salt-tolerant cultivars is still believed to be the most cost-effective strategy to deal with the influence of salt stress on crops. However, the limited resources of salt-tolerant genes restrict the breeding of new tomato varieties. In our study, by comparative proteomic analysis, we found that the advantages of Salt-tolerant tomato in response to salt stress were mainly high level of N- and ABA-related biosynthesis, effective photosynthesis system and active ROS scavenging system, leading a strong tolerance of tomato to salt stress. Additionally, high level of GABA and Brassinolides biosynthesis also contribute the strong tolerance of tomato, and exogenous application of both metabolites can be applied in yield. In parallel, three genes playing important role in salt-resistance of Salt-tolerance tomato were further identified, including A0A3Q7J2V8, A0A3Q7IRF5 and A0A0C6G3Q8, which can be used in future. Additionally, GABA and Brassinolides were also involved in the plant resistance to salt stress. In summary, our results provide mechanistic insights into the advantages of salt-tolerant cultivars vs salt-sensitive cultivars via comparative proteomic analysis.

Conclusions

Our results suggested that salt-tolerant cultivars have more advantages in resistance to salt stress than that of salt-sensitive cultivars. The abundance and activity of antioxidant-related proteins, such as SOD, PPO and POD, were higher in the salt-tolerant cultivars, leading effective advantages in dealing with reactive oxygen species caused by salt stress. Meanwhile, higher expression level of proteins relevant to nitrate and amino acids metabolisms and photosynthetic activity could guarantee a sufficient nutrient for the growth of salt-tolerant under salt stress. Three key proteins functioned in Brassinolides and GABA biosynthesis, including sterol side chain reductase, gamma aminobutyrate transaminase and Starch synthase performed important roles in regulation and activation of salt-resistance of salt-tolerant cultivars.

Supplemental Information

Supplemental Information 1 Supplementary Tables.

Click here for additional data file.

We would like to thank PTM-Biolabs Co., Ltd. (Hangzhou, China) for the quantitative proteomic analysis assistance.

Additional Information and Declarations

Competing Interests

Author Contributions

Data Availability

The authors declare that they have no competing interests.

Qiang Wang conceived and designed the experiments, performed the experiments, analyzed the data, prepared figures and/or tables, authored or reviewed drafts of the paper, and approved the final draft.

Baike Wang performed the experiments, prepared figures and/or tables, and approved the final draft.

Huifang Liu performed the experiments, prepared figures and/or tables, and approved the final draft.

Hongwei Han performed the experiments, authored or reviewed drafts of the paper, and approved the final draft.

Hongmei Zhuang analyzed the data, authored or reviewed drafts of the paper, and approved the final draft.

Juan Wang analyzed the data, prepared figures and/or tables, and approved the final draft.

Tao Yang performed the experiments, authored or reviewed drafts of the paper, and approved the final draft.

Hao Wang conceived and designed the experiments, authored or reviewed drafts of the paper, and approved the final draft.

Yong Qin conceived and designed the experiments, prepared figures and/or tables, and approved the final draft.

The following information was supplied regarding data availability:

The raw measurements are available in the Supplemental Files.

The datasets generated for this study is available at ProteomeXchange (https://www.iprox.org): PXD029130.

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
