# Peer review of "Comparative proteomic analysis for revealing the advantage mechanisms of salt-tolerant tomato (Solanum lycoperscium)"

_PeerJ, doi:10.7717/peerj.12955_

## Round 0.1 · original submission · Major Revisions

Dear Dr. Wang,

Thank you for your submission to PeerJ.

Your article - Comparative proteomic analysis for revealing the advantage mechanisms of salt-tolerant tomato (Solanum lycoperscium) - has been now reviewed by 3 experts, and it requires a number of Major Revisions.

Mainly reviewer #3 raised a number of important concerns regarding mostly the results and discussion sections.

We are willing to receive a new version of your manuscript, given that you are able to address all points raised by the reviewers.

Please address these changes and resubmit. Although not a hard deadline please try to submit your revision within the next 35 days.

Best wishes,

Fabio Nogueira
Academic Editor

Reviewer 1 ·

Basic reporting

The article includes sufficient introduction and background information. It fits into the scope of Peer J. All standard sections are included according to the format required by the journal. It is self-contained and raw data are provided.

Experimental design

Several method sections do not have sufficient details. For example, the reviewer could not evaluate the results of the enzymatic assays because there is no description about sample preparation.

Validity of the findings

Thank authors for providing the raw data for Figures 1, 8 and 9. Please provide more description about the statistical analysis, eg. the number of replicates and how the p values are calculated.

Additional comments

Salinity is a major stressful condition that causes a significant decrease in crop production worldwide. In response, plants have evolved complex salt-responsive signaling which involves significant proteome remodeling processes at the cellular level. Wang et al. analyze how global proteome abundance vary across the salt-tolerant and salt-resistant tomato cultivars and relate the protein expression profile changes to cellular metabolism. They report the following main findings: 1) significant increase in expression of proteins involved in Brassinolides and GABA biosynthesis; 2) higher expression level of antioxidant-related proteins; 3) overexpression of proteins functioned in chloroplast; 4) identification of three key proteins as important salt-resistant resources for breeding salt-tolerant cultivars. In general, the introduction section is well written with recent literature and the proteomic workflow is well established with qRT-PCR verification of gene expression. However, significant modifications of several sections are needed, especially that several method sections do not have sufficient detail and information to replicate.

Major comments
1.Root is the main organ for carrying water and mineral nutrients to the rest of the plant and the primary site of perception and injury for salt stress. Accordingly, root proteome variations in tomato have been a target for study of the molecular mechanism underlying plant salt stress tolerance and adaptation. Examples are from Manaa et al (J Exp Bot. 2011 May;62(8):2797-813) and Gong et al (Biochem Biophys Res Commun. 2014 Mar 28;446(1):417-22). It would be beneficial if the authors would expand the introduction section by explaining the choice of four true-leaf as target of research. Specifically, the authors may compare the effects of salt stress on tomato seedlings (OMICS. 2011 Nov;15(11):801-9), root, leaf, as well as on fruit (OMICS. 2013 Jun;17(6):338-52) to provide a better picture of the proteomic alterations of tomato upon change in salinity.
2.Quantitative LC-MS/MS based proteomic workflow in this study represents a significant technical advantage over previous reports. However, experimental details are needed for the Sample preparation section and the Protein Identification section. Given that the authors are focusing on identifying key proteins as important salt-resistant resources for breeding salt-tolerant cultivars, I believe special attention should be paid to the TMT labeling procedure to minimize potential artifacts. No description of the TMT labeling procedure and corresponding data analysis in the current version of the manuscript.
3.Related to Point #2, in section Sample preparation for TMT proteomic quantification. It has long been recognized that protein carbamylation occurs in urea-containing buffer above 37 degree celsius. Even on ice, sonication can increase local temperature in protein sample, thus significantly enhancing the undesired carbamylation on N-termini of proteins/peptides and at the side chain amino groups of lysine and arginine residues, which may blocks protease digestion and affects protein identification and quantification in mass spectrometry analysis by blocking peptide amino groups from reacting with the amine-reactive NHS ester group on TMT label. It would be beneficial if the authors would provide Venn diagrams of identified proteins among technical replicates for each sample treatment.
4.Some arguments in the Results need clarification since they do not match the data in figures. For example, Lines 192-194 read “Remarkably, the content of H2O2 displayed a lower basal level in Salt-tolerant tomato compared with the Salt-sensitive tomato (Fig.1B)”, however no difference is observed between bar 1 and 3 on Figure 1B. And the following sentence “One hour after imposing, the content of H2O2 in Salt-sensitive tomato was still significantly higher than that in Salt-tolerant cultivars (Fig.1B), which may cause damage to these cultivars.” It is not clear from the Figure 1B what data groups the P values are referring to.

Other comments
5.It is impressive that 7663 proteins are identified using TMT proteomics and, of which, 6501 proteins are quantifiable. It is not clear whether all of the 6501 proteins are quantifiable in all 18 mass spec replicates. Thus it is less obvious that for the section starting from Line 222 (Proteins located at chloroplast exhibited significant difference between SS and ST), are the authors trying to draw a baseline of the proteome differences between SS and ST at time 0 (before treatment)? If so, please make it clear in the text.
6.Lines 118-120: please explain what are the concentrations of salt stress applied and the reason for choosing 200 mmol/L NaCl as final experimental condition.
7.Lines 126-130: please provide the experimental procedures briefly instead of just referring to the assay kits.
8.Lines 131-134: please provide the experimental procedures briefly instead of just referring to the assay kits.
9.Line 139: what is the final conc. of urea after adding trypsin? Trypsin should not work in 8M urea.
10.Line 139: please provide details for peptide extraction and lyophilization.
11.Line 144: Is it a C18 column?
12.STRING database is essentially a literature data-mining resource, which not only contains physical protein-protein interaction data, but also contains functional data. Thus is is crucial to provide the settings when retrieving data from STRING.
13.Line 173: there is no description in the main text whether the three key proteins identified are autophagy-related.
14.Line 253 and Line 270: citations should follow same scheme. Eg. MSANNE et al should be Msanne et al.
15.Line 271: duplicate word “important”.
16.Line 293: “Superoxide (Q7XAV2)” should be “Superoxide dismutase (Q7XAV2)”
17.Line 298: Response should be Responses
18.In Discussion, I feel the tone should be softer in several cases, such as the words on Line 382 “proved”, on Line 395 “guarantee”.
19.Line 502: Brassinolides were also … (missing sentences)
20.Lines 505-506: This sentence is not a good summary since salt-tolerant cultivars, by definition, have more advantage in resistance to salt stress. It would be more appropriate if the authors would change this sentence to similar words like “our results provide mechanistic insights into the advantages of salt-tolerant cultivars vs. salt-sensitive cultivars via comparative proteomic analysis”.
21.Figure 1: Please spell out “ST and SS cultivars” as “salt-tolerant and salt-sensitive cultivars” in the title.
22.Figure 1: Please provide number of replicates used in the figure.
23.Figure 1: “SS0” and “SS1” et al. is confusing without reading the manuscript in detail. Thus I suggest to change them to “SS 0h” and “SS 1h”.
24.Figure 8C-E: please also provide the name of the protein of interest.
25.Figure 8B: Please considering removal of the names for the nodes other than the three proteins of interest for clarity.

Reviewer 2 ·

Basic reporting

1.1 In this manuscript, scattered throughout are words used incorrectly that detract from the reading. Furthermore, I found several subject/verb agreement erros.

For example:

tomato growing ´was` limited by various factors relevant to salt stress` (L 65-66)

mechanisms of tomato resistance to salt stress ´was` still unclear (L 31-32)

Meanwhile, the repair and reconstruction of cell membrane system of seeds ´was` also impaired under saline-alkali stress (L 80-81)

The membrane ion selective absorption capacity ´was` decreased under salt stress (L 82)

Well studied is that salt stress (L 86)

The accumulation of Na+ in soils will result in osmotic stress in plant cells ´will` result in the destruction` (L 68-69)

salt stress ´will` significantly influence the structure of plasma membrane` (L 85)

an in-depth proteome ´were` performed (L 201-202)

only ´protein` in which the log1.5(fold change) > 1 or < -1 and the P 224 value < 0.05 ´were` identified as differentially expressed proteins (L 223-224)


These are mostly things and subject/verb agreement erros that detract from the reading of the manuscript. Thus, I suggest the authors read the manuscript carefully, sentence by sentence, to eliminate grammatical and stylistic errors. I also recommend the authors identify and fix all the subject/verb agreement erros.


1.2 In the introduction, I suggest the authors also cite recent articles (e.g., from 2015 on). Regarding the sentence below, recent studies should also be cited:
´Currently, various mechanisms of plant tolerance to salt stress were elucidated, including reprogramming the photosynthetic pathway (Allakhverdiev et al. 2002; Mittal et al. 2012),enhancing the antioxidant protection ability (Gupta et al. 1993) and synthesis and accumulation of osmotic regulating substances (Yang et al. 2007)` (L 89-92).


1.3 Regarding the sentence ´The physiological indexes of tomato plants were measured using four true-leaf at 0, 1, and 12 h after treatment` (L 122-123), I did not find the results concerning 12 h after NaCl treatment in the topic ´Observable physiological changes between Salt tolerant strain (ST) and Salt sensitive (SS) strain`.

Experimental design

2.1 There are other articles available on proteomics of tomato plants under salt stress. For example:

´Identification of Proteins for Salt Tolerance Using a Comparative Proteomics Analysis of Tomato Accessions with Contrasting Salt Tolerance` ( https://doi.org/10.21273/JASHS.138.5.382)

´Proteomic analysis of salt-stress responsive proteins in roots of tomato (Lycopersicon esculentum L.) plants towards silicon efficiency` ( https://doi.org/10.1007/s10725-015-0045-y )

So I recommend explaining the novelty of the manuscript in the introduction part, citing other articles on this topic.

2.2 Justification for application of 200 mM NaCl and why the physiological indexes of tomato plants were measured at 0, 1, and 12 h after treatment is not clear to me. I recommend explaining (and citing references) on why this NaCl concentration and these times of exposure are suitable for examining tomato genotypes for tolerance/sensitivity to salt stress.

The authors observed: ´one hour after imposing, the content of carbohydrate, H2O2 and total protein in both salt-sensitive and salt-tolerant cultivars were determined. The results showed that no significant change was observed in the content of carbohydrate among both cultivars (Fig.1A). ` (L 189-192). Perhaps the authors would see differences in carbohydrate content if a later exposure time was used instead of 1 h after treatment

Validity of the findings

3. A proteomic analysis of tomato plants is reported in this manuscript. However, I am not sure if salt tolerance mechanisms in tomato were unraveled or at least adequately investigated.

3.1 The authors should provide more details about the tomato cultivars (in the introduction and methods), rather than just mentioning 'tolerant' or 'sensitive'. How did the authors conclude that one tomato cultivar is salt stress-sensitive and another one is tolerant? Have previous studies used these cultivars? Was plant growth of these cultivars under salt-stress conditions previously studied? Once again: I believe that these details should be presented in the introduction or in the methods in order to clarify the contrasting salinity tolerance between cultivars.

3.1.1 Quantifying the effects of salt stress on plant growth is crucial to asserting about the contrasting tolerance. On this topic, I recommend reading the article ´Evaluating physiological responses of plants to salinity stress` ( https://doi.org/10.1093/aob /mcw191). Thus, the authors should present phenotypic and plant growth differences between the tomato genotypes before asserting on contrasting salt tolerance. Please see the following article as another reference on this topic: ´Proteomic Analysis of Seedling Roots of Two Maize Inbred Lines That Differ Significantly in the Salt Stress Response` ( https://doi.org/10.1371/journal.pone.0116697 ).


3.2 Regarding the results on proteomics and antioxidants (figure 03), I recommend the authors present results from control tomato plants (NaCl exposure-free condition) in order to compare salt stress responses that are induced within each genotype, better understand the contrasting salt tolerance, and provide enough information on salt stress-induced responses.

Additional comments

4. Has the proteomic dataset been deposited in any publicly available repository?

Reviewer 3 ·

Basic reporting

Dear Editor,


The review of the manuscript "Comparative proteomic analysis for revealing the advantage mechanisms of salt-tolerant tomato (Solanum lycoperscium)” was completed.

The manuscript possesses major flaws that avoid its publication at the current stage. For instance:

1- ALL TEXT
Authors should carefully check the text of files that were subjected to PeerJ in order to correct typographical mistakes [which could be noticed even in the title (“lycoperscium”), in the first lines of Introduction (“worlf” – line 58), as well as in Supplemental files (“Descrption” – Supplementary Table 2)]. There are glued words too.

INTRODUCTION
2- L68
Please, insert the citation of following statement “The major harmful ions in saline and alkaline soils is Na+”.
What is the source of the Na ions in these soils?

3- Only works from 1985-2011 were cited in the “Introduction” section. Thus, authors should include citations from recent publications in the first part of the manuscript.

MATERIALS AND METHODS
4- L113
4.1- What were the tomato cultivars? Please, insert the names of these cultivar in the M&M.
4.2- What of them is tolerant cultivar to salinity? And the sensitive one?
4.3.- Based on which (previous) work or assays, authors identified these cultivar as tolerant/sensitive to salinity?

5- L114
There are information missing in “peat, perlite and vermiculite”, since authors cited 3 substrates and described the proportion of only two of them “(1:1)”. In addition, what is the unity of such proportion? volume(v), mass (g)? Authors should correct these mistakes too.

6- L122
Based on which works, did authors selected the dose (200 mmol·L-1 NaCl) and time of exposure to stress plants?

7- What was the experimental design?

8- Did authors performed analysis of variance (ANOVA) for evaluation of differences among treatments?
8.1- If so, were data in accordance to ANOVA's assumptions (normal distribution, error independence, and variance homogeneity)?

8.2- Were data subjected to transformation and/or exclusion before ANOVA?

8.3- According to the manuscript, a two-way ANOVA is preferred over the one-way ANOVA, since two factors (i) “tomato cultivars”, and (ii) “salinity levels” were tested.

8.4- What was the p-value used as cut-off in ANOVA?

8.5- What was the value of significance used for differences among treatments in LSD test?

8.6- What test comparison of treatments’ means was used? LSD, Tukey etc? What was the value for significance?

9- The number of biological replications is very low, especially for the biochemical data (SOD, POD etc), so that the freedom degree is unsuitable.

10- Please, there is missing information about PCA in M&M. Please, cite that and other missing information about the statistical procedures in M&M.

RESULTS

11- The “Results” section is very long. For instance, the text of the first subsection can be easily rewritten in 5 lines in one or two setences.

12- L189
It is not usual to use strain in this cases; therefore, replace “strain” by “cultivar”.

In all Figures and Tables
13- Include the meaning of each abbreviation that were included in the Figures and Tables (for instance, “PPO”, “SS”, “ST” etc).

14- Include the number of replications (n) that were used in Figures and Tables?

DISCUSSION
15- This section is very long. In addition to reduce it, authors should separate “Discussion” into subsections according to the subject.

CONCLUSIONS
16- Please, exclude statements like “Our results suggested that salt-tolerant cultivars have more advantages in resistance to salt stress than that of salt-sensitive cultivars.”

Experimental design

See comments above.

Validity of the findings

No comment.

---

## Round 0.2 · Minor Revisions

Dear Dr. Wang,

Thank you for your submission to PeerJ.

I am writing to inform you that your manuscript - Comparative proteomic analysis for revealing the advantage mechanisms of salt-tolerant tomato (Solanum lycoperscium) - has been considered for publication in PeerJ.

However, to fully consider your manuscript for publication in PeerJ, it still requires a few minor revisions.

The suggested changes and reviewer #1 comments are shown below and on your article 'Overview' screen. Please address these changes and resubmit. Please try to submit your revision within the next 15 days.

With kind regards,
Fabio Nogueira
Academic Editor, PeerJ

Reviewer 1 ·

Basic reporting

The manuscript has been improved by authors. Thank authors for addressing my questions. There are still some typos. For example:
1) in Abstract: “were investigated to deciphered” should be “…to decipher”;
2) L224. Proteome cannot be "performed". Please change to "Mass spectroscopy based proteomics was performed.
3) L510 limit
4) Table 1, what is the meaning of "b" in "94.99±3.15b" ?

Experimental design

With the revision, methods are now described with sufficient detail & information to replicate.

Validity of the findings

All sample collection and preparation procedures were performed in triplicate, and each independent biological experiment has 3 technological replicates, Please show the type of the error bar (S.D. vs S.E.M) used in Figures 1, 3, 8, and 9.

Reviewer 2 ·

Basic reporting

No comment.

Experimental design

No comment.

Validity of the findings

No comment.

Additional comments

The authors have considered all my comments. I can see the substantial improvements that have been done in the manuscript. I have no more comments and recommend acceptance.

---

## Round 0.3 · accepted · Accept

Dear Dr. Wang,
Thank you for your submission to PeerJ.
I am writing to inform you that your manuscript - Comparative proteomic analysis for revealing the advantage mechanisms of salt-tolerant tomato (Solanum lycoperscium) - has been Accepted for publication. Congratulations!

Next steps: Your article is being checked and you will receive a list of production tasks shortly. After you complete these tasks, your proofing PDF will be created (please do not proof check your reviewing PDF!).